# Rethinking Sparse Autoencoders: Select-and-Project for Fairness and Control from Encoder Features Alone

Antonio Bărbălau[1]*     Cristian Daniel Păduraru[1]*     Teodor Poncu[2]

Alexandru Țifrea[3]     Elena Burceanu[1]

[1]Bitdefender, Romania [2]University Politehnica of Bucharest, Romania [3]ETH Zurich, Switzerland
{ext-abarbalau,cpaduraru,eburceanu}@bitdefender.com
dan_teodor.poncu@upb.ro, alexandru.tifrea@ethz.ch

## Abstract

Sparse Autoencoders (SAEs) have proven valuable due to their ability to provide interpretable and steerable representations. Current debiasing methods based on SAEs manipulate these sparse activations presuming that feature representations are housed within decoder weights. We challenge this fundamental assumption and introduce an encoder-focused alternative for representation debiasing, contributing three key findings: (i) we highlight an unconventional SAE feature selection strategy, (ii) we propose a novel SAE debiasing methodology that orthogonalizes input embeddings against encoder weights, and (iii) we establish a performance-preserving mechanism during debiasing through encoder weight interpolation. Our **S**election and **P**rojection framework, termed **S&P TopK**, surpasses conventional SAE usage in fairness metrics by a factor of up to $3.2$ and advances state-of-the-art test-time VLM debiasing results by a factor of up to $1.8$ while maintaining downstream performance.

## 1 Introduction

Sparse Autoencoders (SAEs) have become pivotal in mechanistic interpretability through their ability to factorize neural network representations into interpretable components [10, 14, 17]. These sparse decompositions are commonly employed for model steering [1, 6], especially in debiasing contexts where researchers conventionally zero out specific activations associated with unwanted features. Given that this masking operation yields a weighted combination of decoder weights, the prevailing assumption posits that SAE's semantic features are stored within the decoder.

Questioning this assumption, we introduce a SAE-based, encoder-centric debiasing framework. Our methodology, illustrated in Figure 1 and elaborated in Section 3, follows a three-stage process: after computing SAE preactivations, we (i) employ a selection mechanism to identify relevant features, (ii) calculate a weighted sum of the encoder weights corresponding to the selected features to derive a unified bias axis, and (iii) compute a projection that orthogonalizes input vectors relative to this identified axis. We term this feature **S**election and **P**rojection methodology **S&P TopK**, reflecting its use of the top-k features that encode a desired protected attribute. The resulting projection can be applied to debias any input with respect to the specified attribute for any given downstream application. The main contributions embedded within our approach can be summarized as follows:

**1. Challenging the conventional use of SAEs.** In lieu of masking protected SAE attributes, under the assumption that feature representations are stored in decoder weights, we propose orthogonalizing input embeddings with respect to encoder weights.

---

*Equal contribution.

39th Conference on Neural Information Processing Systems (NeurIPS 2025) Workshop: Reliable ML from Unreliable Data.

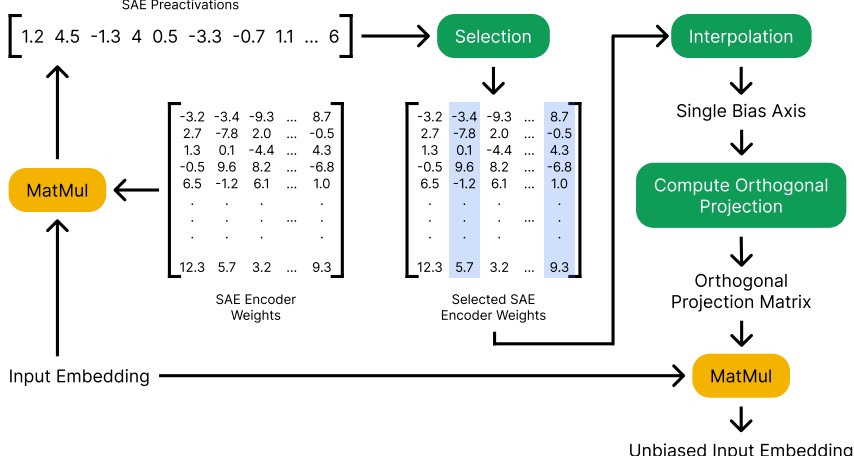

Figure 1: Illustration of the proposed S&P TopK protocol. The main steps of our approach are highlighted in green. We first employ a selection mechanism to identify relevant SAE features. We further propose a debiasing procedure based on orthogonalizing input embeddings with respect to encoder weights. To this end, we compute in the second step a weighted sum of the encoder weights corresponding to the selected features to derive a unified bias axis. Finally, we compute a projection that orthogonalizes input vectors relative to this identified axis.

**2. Highlighting an atypical approach to feature selection.** Our investigation showed that natural approaches to identifying relevant SAE features, such as relying on CLIP scores or training linear probes, are not always optimal, and we highlight Stylist [26] as a robust alternative.

**3. Proposing a mechanism designed to maintain downstream performance during debiasing.** While latent feature masking and orthogonal projections based on SAE features affect the downstream performance, we show that linearly interpolating encoder weights before computing the orthogonal projection fully preserves downstream performance.

**4. Empirical validation showcasing state-of-the-art test-time debiasing performance.** We conduct a test-time VLM debiasing study on the CelebA and FairFace datasets, showcasing that our approach surpasses conventional SAE usage in terms of inducing fairness, as measured by conventional KL Divergence metrics, improving results by a factor of 3.2 and advances state-of-the-art results by a factor of 1.8 while maintaining downstream performance.

## 2 Problem Setting

We consider the problem of debiasing embeddings obtained with a large pretrained model (e.g. VLM) at test-time in retrieval and classification setups. Test-time VLM debiasing constitutes a research domain focused on eliminating a protected attribute $\mathbf{a}$ (*e.g.*, gender) from VLM (*e.g.*, CLIP[22]) representations, while preserving other attribute information. Under this paradigm, the VLM operates as a black-box system with modifications applied exclusively to final output representations. This setup features a *reference dataset* $\mathcal{R}$ containing images labeled with a desired protected attribute $\mathbf{a}$ (*e.g.*, *male* and *female* annotations) in order to pinpoint it for removal. We designate $\mathcal{R}_{a_i}$ as the reference subset where attribute $\mathbf{a}$ assumes value $a_i$.

Debiasing performance is assesed on retrieval and classification benchmarks and quantified via KL divergence and MaxSkew metrics [11] that compare the distribution of $\mathbf{a}$ in the dataset against its distribution within retrieved results. Furthermore, downstream performance is measured for classification tasks by means of the worst group ROC-AUC (wgROC-AUC) metric [11], where groups are formed by means of combining attributes and labels, e.g. (female, blonde hair) for CelebA.

## 3 Method

Our approach utilizes a JumpReLU SAE [23] trained on VLM image embeddings $x \in \mathbb{R}^n$. The SAE framework incorporates linear encoder and decoder layers with projection matrices $E \in$

Table 1: We present results on CelebA evaluating various combinations of feature selection and removal protocols, while simultaneously demonstrating the effectiveness of our proposed axis interpolation technique. Our findings reveal that interpolation preserves downstream accuracy, Stylist outperforms linear probing as a selection mechanism, and projection against encoder weights substantially exceeds masked reconstruction in terms of debiasing performance.

| Selection | Removal | Interpolation | KL $\downarrow$ | MaxSkew $\downarrow$ | wgROC-AUC $\uparrow$ |
|---|---|---|---|---|---|
| None | None | - | 0.113880 | 0.293723 | 0.754743 |
| CLIP Score | $\perp$ TopK Encoder Weights | - | 0.164876 | 0.308559 | 0.744376 |
| LP | $\perp$ TopK Encoder Weights | - | 0.055613 | 0.250359 | 0.631793 |
| Stylist | $\perp$ TopK Encoder Weights | - | **0.035051** | **0.235039** | 0.629358 |
| Stylist | Masked Reconstruction | N/A | 0.061290 | 0.263063 | 0.527940 |
| Stylist | $\perp$ TopK Decoder Weights | - | 0.067286 | 0.299477 | 0.651578 |
| Stylist | $\perp$ TopK x Weights | $\checkmark$ | 0.079235 | 0.260566 | **0.752426** |

$\mathbb{R}^{n,m}, D \in \mathbb{R}^{m,n}$ and biases $b_E \in \mathbb{R}^m, b_D \in \mathbb{R}^n$. Preactivations within the SAE are expressed as $z = (x - b_D)E + b_E$, leading to activations $\hat{z} = \text{JumpReLU}(z)$ and subsequent reconstructions $\hat{x} = \hat{z}D + b_D$. For selective weight operations, we establish the notation $W_{:,i} = [W_{1,i}W_{2,i} \dots W_{n,i}]^\top$ to represent individual weight columns, and $W_{:,I} = [W_{i_1}W_{i_2} \dots W_{i_q}]$ for index collections $I = (i_1, i_2, \dots i_q)$, thereby enabling precise weight subset selection. Similarly, we use $W_{I,:}$ to denote row selection operations.

**Feature selection.** We forward three approaches aimed at identifying the SAE features that correspond to the protected attribute $\mathbf{a}$: (i) using a linear probe to weight SAE features (ii) using CLIP score to identify which features are correlated with the protected concept and (iii) using Stylist to see which SAE features vary the most across the different reference subsets $R_{a_i}$.

For the initial methodology, we train a Linear Probe (LP) on the reference dataset $\mathcal{R}$ to predict attribute $\mathbf{a}$ using SAE preactivations as input features. Subsequently, we rank SAE features according to the absolute magnitude of their corresponding classifier weights.

For the second methodology, we generate text prompts following the template "a photo of a $a_i$" for each instantiation $a_i$ (*e.g.* , male, female) of the protected attribute $\mathbf{a}$. We subsequently compute CLIP scores between these prompts and image embeddings $x \in \mathcal{R}$ from the reference dataset. SAE features are then ranked according to the correlation between their preactivation values across samples in $\mathcal{R}$ and the corresponding CLIP scores for each sample.

Finally, we highlight the applicability of Stylist [26] within the current context, a technique originally proposed for novelty detection. Stylist ranks features by computing the average Wasserstein distance between the distributions of preactivation values $z_t$ of each feature $t$ across different reference subsets $\mathcal{R}_{a_i}$ and $\mathcal{R}_{a_j}$ where $i \neq j$. Since attribute $\mathbf{a}$ represents the primary source of variation among reference subsets, features exhibiting the largest distributional distances should correspond to encodings of $\mathbf{a}$.

After applying the chosen selection method, we proceed with the top-k SAE features, and denote the selected subset as $S$.

**Synthesizing the protected attribute axis.** We train a Logistic Regression classifier with weights $w \in \mathbb{R}^k$ to predict attribute $\mathbf{a}$ using preactivations from the chosen feature subset $S$. The variation axis $v \in \mathbb{R}^n$ for attribute $\mathbf{a}$ is subsequently constructed through a weighted aggregation of encoder weights, where the weights correspond to the learned classifier parameters: $v = E_{:,S}w^\top$.

**Orthogonal projection.** We orthogonalize the image embeddings $x$ with respect to the identified axis $v$ through projection using the orthogonal projection matrix $V = \mathcal{I}_n - v(v^\top v)^{-1}v^\top$ [5], where $\mathcal{I}_n$ denotes the $n$-dimensional identity matrix.

## 4   Experimental setup

**Datasets.** We use CelebA [20], which contains over 200,000 images annotated with facial attributes, to analyze gender bias in hair color classification. We evaluate both accuracy and fairness. We use FairFace [16], with over 100,000 demographically balanced images, for fairness evaluation of the stereotype-based retrieval tasks (*e.g. violent person*, *burglar*), that reflect gender bias.

Table 2: CelebA evaluation encompassing multiple state-of-the-art methods, where asterisk-marked (*) results are sourced from [11]. Findings reveal that our approach significantly surpasses the standard SAE debiasing procedure utilizing linear probe-based selection and masked reconstruction removal. Notably, our method helps establish new state-of-the-art results for KL Divergence and MaxSkew when combined with BendVLM.

| Method | Debiases Input | Debiases Prompt | Downstream Knowledge | KL ↓ | MaxSkew ↓ |
|---|---|---|---|---|---|
| Vanilla | - | - | - | .1138 ± .0059 | .2937 ± .0077 |
| Regular SAE (LP & MR) | ✓ | - | - | .2604 ± .1540 | .5735 ± .1790 |
| BendVLM P0 [11] | - | ✓ | - | .1485 ± .0052 | .2915 ± .0178 |
| **S&P TopK** | ✓ | - | - | .0792 ± .0067 | .2605 ± .0148 |
| OrthoProj* [5] | - | ✓ | - | .0710 ± .0030 | .2520 ± .0060 |
| OrthoCali* [5] | - | ✓ | ✓ | .0590 ± .0010 | .2600 ± .0040 |
| BendVLM [11] | - | ✓ | ✓ | .0186 ± .0062 | .1803 ± .0316 |
| **S&P TopK** + BendVLM | ✓ | ✓ | ✓ | **.0101** ± .0044 | **.1153** ± .0266 |

**Models.** We employ `CLIP ViT-B/16` as the target VLM for debiasing. We train the JumpReLU SAE [23] with 16,384 features, following the methodology outlined in [2] on approximately 37M images from CC12M [4], ImageNet-21k [24], ImageNet-1k [7], ImageNet-A [13], ImageNet-R [12], ImageNet-Sketch [28] and a small subset of LAION-2B-en [25].

# 5   Results

We provide extended details about the experimental setup in Appx. E and additional results in Appx. F. We summarize the main takeaways from our experiments as follows:

**Maintaining downstream performance.** As shown in Table 1 and Table 4 from Appx. F, Regular SAE usage via Masked Reconstruction leads to a noticeable drop in wgROC-AUC. In contrast, our projection-based debiasing consistently preserves more performance than masked reconstruction. Finally, and most importantly, we highlight that our proposed interpolation strategy fully preserves downstream task accuracy across all combinations of selection and removal methods.

**Unconventional feature selection.** As shown in Tables 1, 5 and 4 the selection based on CLIP score does not manage to pinpoint relevant features. Furthermore, the selection based on linear probing is not always optimal: on CelebA, when projecting with respect to encoder weights, the selection provided by Stylist yields KL Divergence results which are better by a factor of 1.5.

**Encoder-based debiasing.**. As shown in Tables 1, 5 and 4 computing the projection matrix based on encoder weights rather than decoder weights yields a 1.9x increase in performance, measured in terms of KL Divergence, on CelebA and a 2.5x improvement on FairFace. We further highlight that our proposed debiasing mechanism based on orthogonalizing with respect to encoder weights, outperforms the standard procedure of masked reconstruction, yielding a 1.3x increase in perfromance on FairFace and a 1.7x increase in performance on CelebA.

**State-of-the-art test-time debiasing results.** As shown in Table 2, our method significantly outperforms Regular SAE debiasing with selection based on Linear Probing (LP) and removal via Masked Reconstruction (MR), yielding a 3.2x improvement in KL Divergence. Furthermore, when combined with prompt debiasing techniques, it manages to improve upon the state-of-the-art results, yielding a 1.8x improvement. We observe a similar outcome on the FairFace dataset, as shown in Table 6. Furthermore, unlike BendVLM and OrthoProj, our method does not make use of CLIP's contrastive properties, making it applicable to unimodal and generative models as well.

# 6   Conclusion

We reexamined conventional SAE-based representation debiasing. By exploiting encoder weights through our selection and projection architecture, complemented by interpolation, our **S&P TopK** approach realizes substantial fairness enhancements without compromising task utility. Results on CelebA and FairFace establish new state-of-the-art performance in test-time VLM debiasing.

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

## Appendix

## A  Limitations

Our methodology requires a SAE trained on image embeddings of the VLM to be debiased. This implies that a new SAE must be trained for each new VLM that one wants to debias, leading to increased costs. The current formulation of the interpolation step only targets binary attributes, but other protected attributes (*e.g.*, race or religion) do not fall in this category. Lastly, the current work also does not address the whac-a-mole dilemma [19], a known phenomena whereby mitigating one bias leads to an amplification of a different bias

## B  Software

The code that reproduces the main experiment can be accessed at the following link.

## C  Related work

### C.1  Sparse Autoencoders

While Sparse Autoencoders present themselves are a remarkable and useful approach to model steering and interpretability, there has been a recent wave of pessimism in the literature. A recent systematic evaluation [27] shows that SAEs perform worse than linear probes on an out-of-distribution harmful-intent detection task. Similar negative results have appeared for interpretability, unlearning, steering, robustness [9, 15, 21]. Kantamneni et al. [15] found that SAE probes fail to offer a consistent overall advantage when added to a simulated practitioner's toolkit. Mayne et al. [21] analyzed the use of SAEs for interpreting steering vectors finding that (i) steering vectors fall outside the input distribution for which SAEs are designed, and (ii) steering vectors can have meaningful negative projections in SAE feature directions, which SAEs are not designed to accommodate. Farell et al.[9] found that "zero ablating features is ineffective" and that simultaneous interventions across multiple SAE features, while capable of unlearning various topics, produce comparable or greater unwanted side effects than existing techniques. These findings suggest that substantial improvements in either SAE quality or intervention methodologies are necessary. Through our work we aim to forward a new perspective upon SAE usage which may alleviate some of the existing pessimism.

### C.2  Test-time Debiasing

Berg et al. [3] propose a VLM debiasing method that adds a trainable soft prefix to textual prompts in order to suppress the protected attribute. The soft prefix is trained such that it only suppresses the attribute in prompts that do not explicitly feature said attribute, maintaining the image-text alignment in such situations. This is achieved through a mixture of the original CLIP [22] loss and an adversarial loss that prevents an MLP from predicting the protected attribute of an image based on its CLIP scores with respect to prompts that do not feature the attribute.

Chuang et al. [5] introduce two debiasing methods, dubbed **OrthProj** and **OrthCali**. In OrthProj they make the query embeddings orthogonal to text embeddings of prompts featuring only instances of the protected attribute. OrthCali starts from the projection matrix of OrthProj and calibrates it such that it also minimizes the post-projection distance between embeddings of prompt-pairs that feature the attribute of interest but differ only in the value of $\mathbf{a}$, the protected one (*e.g.*, 'a photo of a male doctor' and 'a photo of a female doctor').

**BendVLM** [11] is a state-of-the-art two-stage debiasing method that uses additional information from the downstream task. For a given retrieval prompt (*e.g.*, "a photo of a doctor") it estimates a local protected attribute axis from embeddings of prompts featuring both the protected (*gender*) and target (*doctor*) attributes. It then optimizes the text embedding to be equidistant from a set of reference image embeddings that feature the target attribute but differ in value of the protected attribute.

Table 3: Comparison of our method and the CAV baseline on the CelebA dataset.

| Method | KL ↓ | MaxSkew ↓ | wgROC-AUC ↑ |
|--------|------|-----------|-------------|
| Vanilla | 0.113880 | 0.293723 | 0.754743 |
| CAV | 0.145891 | 0.288400 | 0.754424 |
| S&P TopK | 0.079235 | 0.260566 | 0.752426 |

## D   Intuition

Our work is motivated by the conceptual similarity between Concept Activation Vectors (CAVs) [18] and SAE encoder weights. CAVs represent directional vectors pointing toward samples containing the concept of interest and away from those lacking it. We observe analogous behavior in SAE encoder weights, which function as attribute detectors. For feature activation to occur, the corresponding encoder weight must exhibit positive cosine similarity with samples containing the target attribute and negative similarity with samples lacking it (since preactivation $z_i$ can be expressed as $\cos(x, E_{:,i}) \|x\|_2 \|E_{:,i}\|_2$, representing the cosine similarity scaled by vector norms).

In our application, we seek features corresponding to concepts like 'male' or 'female'. However, SAE features do not encode pure 'male' or 'female' attributes, but rather composite representations such as 'human + male' and 'human + female'. These features consequently capture human characteristics (*e.g.*, hair, eyes) alongside gender information. Direct projection onto existing encoder features removes not only gender concepts but also essential human traits like hair-related features, explaining performance degradation on CelebA. Our interpolation approach using linear classifier weights effectively computes the difference between 'human + male' and 'human + female' features by assigning positive weights to one gender's features and negative weights to the other, thereby eliminating the shared 'human' component and yielding a 'male - female' variation axis that preserves task-relevant information during projection.

The interpolated SAE encoder axis outperforms regular CAVs trained on image embeddings due to several key factors. Since interpolation weights $w$ are trained on SAE preactivations, the operation $(xE_{:,S})w$ can be regrouped as $x(E_{:,S}w)$. With $u = E_{:,S}w$ representing a vector in $\mathbb{R}^n$, we effectively learn a vector using the same data as the CAV baseline in Table 3. The crucial difference is that $u$ is constrained to a lower-dimensional subspace defined by the span of columns in $E_{:,S}$. This constraint prevents $u$ from exploiting spurious correlations for classification, as it can only utilize concepts encoded by the selected features. Consequently, with proper feature set $S$ selection, a CAV for the target attribute can be learned even from noisy data containing spurious correlations, and with fewer examples due to the reduced parameter count compared to regular CAVs ($k \ll n$).

## E   Experimental Setup Details

We engage two lines of experimentation in the context of the current work.

**Debiasing with Sparse Autoencoders.** Our first experimental line investigates optimal sparse autoencoder utilization for test-time debiasing. Results are presented in Tables 1, 5 and 4. We evaluate three SAE feature selection methods: CLIP score correlation (CLIP Score), linear probing (LP), and Stylist. Additionally, we assess different debiasing approaches: Masked Reconstruction, orthogonal projection against encoder weights ("⊥ TopK Encoder Weights"), and orthogonal projection against decoder weights ("⊥ TopK Decoder Weights"). We also validate our proposed weight interpolation technique for preserving downstream performance.

In Tables 1 and 5, we initially assess selection methods using a fixed removal approach without weight interpolation. After identifying the optimal selection method, we present results across different removal techniques. Table 4 provides comprehensive results covering all combinations of selection, removal, and interpolation choices.

**Masked Reconstruction baseline.** To account for the inherent reconstruction error of SAEs [8] we follow standard procedure and subtract from the original input the reconstruction of the selected features. That is, the final debiased input is $x - \hat{z}_{:,S}D_{S,:}$.

Table 4: We present results on CelebA evaluating all combinations of feature selection and removal protocols, while simultaneously demonstrating the effectiveness of our proposed axis interpolation technique.

| Selection | Removal | Interpolation | KL ↓ | MaxSkew ↓ | wgROC-AUC ↑ |
|-----------|---------|---------------|------|-----------|-------------|
| None | None | - | 0.113880 | 0.293723 | 0.754743 |
| CLIP Score | Masked Reconstruction | N/A | 0.101393 | **0.237183** | 0.747717 |
| CLIP Score | ⊥ TopK Decoder Weights | - | 0.096210 | 0.305892 | 0.750047 |
| CLIP Score | ⊥ TopK Encoder Weights | - | 0.164876 | 0.308559 | 0.744376 |
| CLIP Score | ⊥ TopK Decoder Weights | ✓ | 0.130474 | 0.332680 | **0.755762** |
| CLIP Score | ⊥ TopK Encoder Weights | ✓ | 0.122708 | 0.317317 | **0.753262** |
| LP | Masked Reconstruction | N/A | 0.260455 | 0.573577 | 0.521133 |
| LP | ⊥ TopK Decoder Weights | - | 0.083154 | 0.319729 | 0.654926 |
| LP | ⊥ TopK Encoder Weights | - | **0.055613** | **0.250359** | 0.631793 |
| LP | ⊥ TopK Decoder Weights | ✓ | 0.103445 | 0.275211 | 0.753322 |
| LP | ⊥ TopK Encoder Weights | ✓ | 0.104126 | 0.288260 | 0.751229 |
| Stylist | Masked Reconstruction | N/A | **0.061290** | 0.263063 | 0.527940 |
| Stylist | ⊥ TopK Decoder Weights | - | 0.067286 | 0.299477 | 0.651578 |
| Stylist | ⊥ TopK Encoder Weights | - | **0.035051** | **0.235039** | 0.629358 |
| Stylist | ⊥ TopK Decoder Weights | ✓ | 0.098754 | 0.317625 | 0.751755 |
| Stylist | ⊥ TopK Encoder Weights | ✓ | 0.079235 | 0.260566 | **0.752426** |

**Comparison with state-of-the-art results.** Our second experimental line compares our method against existing state-of-the-art approaches in Tables 2 and 6. We evaluate against OrthoProj and OrthoCali [5], both the "P0" projection component and complete BendVLM method [11], and a standard SAE debiasing protocol using Linear Probing (LP) for selection and Mask Reconstruction (MR) for removal.

We distinguish between debiasing approaches based on their intervention targets. Our method operates on input images, while others like BendVLM debias CLIP prompts used for retrieval. During calibration, both OrthoCali and BendVLM leverage downstream task prompts for more informed debiasing.

We also evaluate our method combined with BendVLM, as they complement each other by debiasing input embeddings and retrieval prompts respectively.

**Tasks.** On both datasets we debias the image embeddings with respect to the 'gender' attribute. We note that the FairFace [16] dataset is only annotated for 'race', 'gender' and 'age', which constitute protected attributes. As such there is no annotated downstream task on which the wgROC-AUC performance metric can be reported, as opposed to the CelebA [20] dataset where the 'hair-color' attribute is used.

**Implementation details.** We consistently set $k = 16$ throughout our experimental evaluation. Linear probes are implemented as Logistic Regressors featuring an L2 penalty, no bias, and class balancing weights. For all experiments we followed the setup proposed by BendVLM [11], which implies a 5-fold validation using 50% of the samples as a reference dataset. The KL divergence and MaxSkew metrics are computed using the top 500 retrieved samples. Consequently, in Tables 1 and 5 we report the mean and the confidence intervals for all methods.

## F Additional Results

We present additional results on the FairFace dataset in Tables 5 and 6, along with comprehensive results covering all selection, removal, and interpolation combinations in Table 4. These findings reinforce the conclusions outlined in Section 5. Notably, linear probing yields optimal SAE feature selection for FairFace, demonstrating that no universal best method exists for feature identification. However, Stylist achieves comparable performance and exhibits greater overall robustness across datasets.

Table 5: We present results on FairFace evaluating various combinations of feature selection and removal protocols, while simultaneously demonstrating the effectiveness of our proposed axis interpolation technique. Our findings reveal that linear probing outperforms Stylist as a selection mechanism on this dataset, and that projection against encoder weights still exceeds masked reconstruction in terms of debiasing performance.

| Selection | Removal | KL $\downarrow$ | MaxSkew $\downarrow$ |
|---|---|---|---|
| None | None | 0.129757 | 0.334185 |
| CLIP Score | $\perp$ TopK Encoder Weights | 0.346062 | 0.560762 |
| LP | $\perp$ TopK Encoder Weights | **0.041860** | **0.195931** |
| Stylist | $\perp$ TopK Encoder Weights | 0.047666 | 0.204429 |
| LP | Masked Reconstruction | 0.057230 | 0.224937 |
| LP | $\perp$ TopK Decoder Weights | 0.105178 | 0.325300 |

Table 6: FairFace evaluation encompassing multiple state-of-the-art methods, where asterisk-marked (*) results are sourced from [11]. Findings reveal that our approach surpasses the standard SAE debiasing procedure utilizing linear probe-based selection and masked reconstruction removal. Notably, our method helps establish new state-of-the-art results for KL Divergence and MaxSkew when combined with BendVLM.

| Method | Debiases Input | Debiases Prompt | Downstream Knowledge | KL $\downarrow$ | MaxSkew $\downarrow$ |
|---|---|---|---|---|---|
| Vanilla | - | - | - | $.1297 \pm .0025$ | $.3341 \pm .0056$ |
| Regular SAE | $\checkmark$ | - | - | $.0572 \pm .0147$ | $.2249 \pm .0296$ |
| BendVLM P0 | - | $\checkmark$ | - | $.3283 \pm .0038$ | $.5147 \pm .0060$ |
| **S&P TopK** | $\checkmark$ | - | - | $.0476 \pm .0062$ | $.2044 \pm .0157$ |
| OrthoProj* | - | $\checkmark$ | - | $.3400 \pm .0030$ | $.5200 \pm .0010$ |
| OrthoCali* | - | $\checkmark$ | $\checkmark$ | $.4260 \pm .0020$ | $.6060 \pm .0010$ |
| BendVLM | - | $\checkmark$ | $\checkmark$ | $.0100 \pm .0016$ | $.1166 \pm .0101$ |
| **S&P TopK** + BendVLM | $\checkmark$ | $\checkmark$ | $\checkmark$ | $\mathbf{.0080} \pm .0029$ | $\mathbf{.1001} \pm .0241$ |

