# OpenReview forum: "Rethinking Sparse Autoencoders: Select-and-Project for Fairness and Control from Encoder Features Alone"
_NeurIPS.cc/2025/Workshop/Reliable_ML — NeurIPS 2025 - Reliable ML Workshop_

### Official Review · Reviewer_TTkU · 2025-09-17
**Comment**

**Rating:** 7
**Confidence:** 3

**Review:**

The paper proposes an encoder based framework for sparse autoencoder debiasing.

Weaknesses, and Suggestions:

1. Figure one does not display correctly.

2. Since paper does not exceed the page limit, it might be a good idea to move supplementary pages to the main. It would be nice to have the intuition section and table 3 included as main.


Strengths:
The paper is well written and the idea is interesting.

---

### Official Review · Reviewer_1Eqa · 2025-09-20
**Review for Rethinking SAE**

**Rating:** 6
**Confidence:** 1

**Review:**

**Summary**

This work proposes an Sparse Autoencoder-based encoder-centric debiasing framework called S&P Top K for debiasing embeddings at test time while preserving task utility.

**Strengths**

1. This work moves beyond fundamental assumption in SAE on *decoder weights hold feature representations* and investigates the encoder-centric perspective with a proof of concept that the proposed method achieved SOTA performance in test-time VLM debiasing, the insight and contribution of which is valuable.
2. The empirical analysis is sound and thorough **with two datasets (tasks).

**Weaknesses**

1. The introduction does not clearly present the gap and problem, which makes it difficult for understanding the proposal, e.g. why proposes an encoder-centric debiasing framework compared to other debiasing methods.